# The Fate of Th17 Cells is Shaped by Epigenetic Modifications and Remodeled by the Tumor Microenvironment

**DOI:** 10.3390/ijms21051673

**Published:** 2020-02-29

**Authors:** Elodie Renaude, Marie Kroemer, Romain Loyon, Delphine Binda, Christophe Borg, Michaël Guittaut, Eric Hervouet, Paul Peixoto

**Affiliations:** 1University of Bourgogne Franche-Comté, INSERM, EFS BFC, UMR1098, Interactions Hôte-Greffon-Tumeur/Ingénierie Cellulaire et Génique, F-25000 Besançon, France; elodie.renaude@edu.univ-fcomte.fr (E.R.); mariekroemer@gmail.com (M.K.); romain.loyon@gmail.com (R.L.); delphine.binda@gmail.com (D.B.); xtoph.borg@gmail.com (C.B.); michael.guittaut@univ-fcomte.fr (M.G.); eric.hervouet@univ-fcomte.fr (E.H.); 2Centre Hospitalier Universitaire de Besançon, Centre d’Investigation Clinique, INSERM CIC 1431, 25030 Besançon, France; 3Department of Pharmacy, University Hospital of Besançon, F-25000 Besançon, France; 4Department of Medical Oncology, University Hospital of Besançon, F-25000 Besançon, France; 5DImaCell Platform, University of Bourgogne Franche-Comté, F-25000 Besançon, France; 6EPIGENEXP Platform, University of Bourgogne Franche-Comté, F-25000 Besançon, France

**Keywords:** epigenetic modifications, Th17 cells, tumor microenvironment, TIL

## Abstract

Th17 cells represent a subset of CD4+ T cells characterized by the master transcription factor RORγt and the production of IL-17. Epigenetic modifications such as post-translational histone modifications and DNA methylation play a key role in Th17 cell differentiation and high plasticity. Th17 cells are highly recruited in many types of cancer and can be associated with good or bad prognosis. Here, we will review the remodeling of the epigenome induced by the tumor microenvironment, which may explain Th17 cell predominance. We will also discuss the promising treatment perspectives of molecules targeting epigenetic enzymes to remodel a Th17-enriched tumor microenvironment.

## 1. Introduction

T helper 17 (Th17) cells are responsible for the protection of epithelial and mucosal barriers against extracellular bacteria and fungi [1]. This subset of CD4+ T cells produces interleukin-17 (IL-17), IL-21, IL-22, and is characterized by the expression of the transcription factor RORγt (retinoic acid receptor–related orphan receptor γt) [2,3]. Other secondary transcription factors such as c-Maf, IRF4 (interferon regulatory factor 4), BATF (basic leucine zipper transcription factor ATF like), and IkBζ, are required for the induction of RORγt and the secretion of specific cytokines. Th17 cells are induced upon T-cell receptor activation in the presence of transforming growth factor TGF-β1 and IL-6. TGF-β activates the Smad signaling pathway which is common to Th17 and regulatory T cells (Treg) whereas IL-6 induces the signal transducer and activator of transcription 3 (STAT3) signaling pathway responsible for the inhibition of Treg development and RORγt expression [4]. IL-23 and IL-21 are required for Th17 cell stability and maturation.

Epigenetic modifications which can be defined as a reversible process that alters gene activity without changing the DNA sequence are involved in CD4 T cell polarization and can explain the great plasticity of the Th17 cell subset [5]. Epigenetic modifications regroup DNA methylation on CpG islands of the promoters and post-translational histone modifications (acetylation, methylation, phosphorylation, ubiquitination of lysine/arginine residues of the histone tails). These modifications, directly or indirectly, regulate chromatin compaction and accessibility to transcription factors. For example, the acetylation of lysine (K) residues on histone proteins by histone acetyltransferases (HATs) results in the suppression of positive charges, thus decreasing their association with DNA and leading to chromatin decompaction favorable to transcription factor recruitment. On the other hand, histone deacetylases (HDACs) induce a strong association of histone proteins with DNA, thus preventing the binding of transcription factors. Unlike histone acetylation, enzymes responsible for histone methylation are site-specific and the methylation marks can result in either a positive or a negative effect on transcription depending on the sites and the number of methylations on histone tails. For instance, the di- or tri-methylation of lysine 4 of histone 3 proteins (H3K4me2/3) is permissive to transcription whereas the di- or tri-methylation of lysine 27 of histone 3 proteins (H3K27me2/3) is a repressive mark. Enzymes which add these modifications are called histone methyltransferases (HMTs) and the removal of these marks is mediated by histone demethylases. Another epigenetic modification is hyper methylation of CpG islands of promoter regions by DNA methyltransferases (DNMTs) leading to transcriptional gene repression. Epigenetic regulation can therefore be summarized with a model comprising “writers”, “readers”, and “erasers” in which the deposition of epigenetic marks by “writers” such as HATs, HMTs, or DNMTs are then recognized by “readers”. These readers are proteins that bind different covalent modifications present on DNA or histones and mediate the recruitment of transcription factors or the intervention of other epigenetic enzymes which activate or repress transcription. Since Epigenetics is a reversible process, erasers like HDACs and histone demethylases can remove epigenetic marks [6]. Epigenetic regulation of gene expression can also occur after transcription with non-coding RNAs, like micro RNAs (miRNAs), that can specifically bind the 3′-UTR sequence of targeted mRNA, thus inhibiting their translation and inducing their degradation [7].

In this paper, we will discuss the implication of epigenetics in the context of anti-tumor immunity. Although it is associated with good or bad prognosis, many human cancer types present intra-tumoral recruitment of Th17 cells compared to healthy tissues, demonstrating the role of the tumor microenvironment on either recruitment, polarization, reprogramming or amplification of these cells. Here, we will also discuss the tumor microenvironment effects on epigenetics responsible for an intra-tumoral Th17 landscape and its consequences on immune evasion, as well as the promising treatment perspectives using molecules targeting epigenetic enzymes.

## 2. Epigenetics Plays a Key Role in Th17 Cells Lineage Commitment and Plasticity

### 2.1. Epigenetic Initiation of the Th17 Differentiation Program

The regulation of gene expression generally involves pioneer transcription factors which induce chromatin remodeling, allowing accessibility of cis regulatory elements to secondary transcription factors. Depending of the cytokinic environment surrounding naive CD4 T cells, different STAT transcription factors can be activated and then can bind to specific loci to regulate the expression of genes involved in T helper differentiation. The STAT transcription factor family is also responsible for the remodeling of the epigenetic landscape [8]. In Th17 cells, the activation of STAT3 by IL-6 positively regulates the permissive marks H3K4me3 on the *Il-17* locus (Figure 1) [9]. 

Upstream STAT3 induction, epigenetic modifications are also involved in Th17 differentiation. Recently, Lin et al. demonstrated that Th17 differentiation depends on an upstream mechanism regulated by epigenetics. By maintaining the permissive mark H3K4me3 on the promoter of the *IL-6 Receptor*, Cxxc1 allows for the expression of the *IL-6 receptor* and enables the IL-6/STAT3 signaling pathway thus regulating the balance between Th17 and regulatory T cells [10]. 

With meta-analysis of multiple RNAseq and transcription factor genome occupancy datasets validated by in vitro experiments, Ciofani et al. proposed a network regulatory model for Th17 lineage commitment. Following TCR activation of CD4 T cells, the transcription factors BATF and IRF4 are transcriptionally induced and then co-localized at key lineage-associated loci (*Il17a*, *Il17f*, *Il12rb1*, *Rorcγt*). The cooperative binding of BATF and IRF4 was associated with chromatin accessibility. This pre-patterning of chromatin allows for further accessibility of specific cis regulatory elements to transcription factors induced by STAT3 activation. The initiation of a specific lineage transcriptional program is then globally tuned by the subsequent recruitment of the master transcription factor RORγt [11]. Downstream of the STAT3 signaling pathway, Jiang et al., demonstrated that the epigenetic regulator Tripartite motif containing 28 (TRIM28) binding was largely co-localized with the active epigenetic marks H3K4Me3 and DNA hydroxyl-methylation (5hmc) at specific Th17 cells related genes (*Il17-Il17f*, *Il-21*, *RORc*, *RORα*, *Batf* and *Irf4*). TRIM28 can recruit different DNA and histone modification enzymes. Protein immunoprecipitation analysis with anti-TRIM28 antibodies revealed that TRIM28 is also involved in a complex including STAT3 and RORγt. A knock-down of the expression of STAT3, IRF4 and BATF and chromatin immunoprecipitation (ChIP) assays with anti-TRIM28 antibodies revealed that the recruitment of TRIM28 at the *Il17-Il17f* locus is dependent of STAT3 and its co-factors IRF4 and BATF but not of RORγt. These data suggested that the epigenetic regulator TRIM28 is first recruited at the *Il17-Il17f* locus and then allows for the binding of RORγt to lead to IL-17 expression [12]. A schematic representation of the epigenetic regulation of *Il-17* expression in Th17 cells is described Figure 1. 

Epigenetic interventions during Th17 differentiation occur at different timelines and are submitted to a complex regulatory network. Several transcription factors have been associated with the deposition of permissive or repressive histone marks at Th17 specific gene loci and are believed to regulate the chromatin state of Th17 lineage-determining genes prior to and after differentiation. However, a direct or complete regulatory mechanism has not been described yet. Another epigenetic regulator of the Th17 initiation program is the transcription factor Ikaros. Indeed, in naive CD4 T cells, Ikaros is required to maintain the possibility of further Th17 differentiation by limiting repressive chromatin modifications at Th17 specific gene loci such as *Ahr*, *Runx1*, *Rorc*, *Il17a*, *Il22*. After cytokine-driven Th17 polarization, chromatin remodeling by Ikaros enabled the acquisition of permissive histone marks at these loci [13]. Other epigenetic mechanisms involving histone H3 lysine-27 demethylase JMJD3 have been found to play a role in Th17 differentiation. Indeed, the knock down of JMJD3 in murine CD4 T cells impaired Th17 differentiation. The level of H3K27me3 at the *Rorγc* regulatory elements is specifically decreased by JMJD3 in Th17 cells. The loss of this repressive histone mark favorably changes the chromatin accessibility of the *Ror*γ*c* locus [14]. Further studies will be needed to clarify how JMJD3 selectively promotes Th17 cell differentiation. Possible interactions of JMJD3 with RORγt and STAT3 which were previously described by Ciofani et al. may be part of this explanation [11]. Furthermore, implication of post translational regulation of Th17 differentiation by miRNA has been reported [15]. For example, in vitro, Th17 cells were found to have higher expression of miR-326 than other CD4 lymphocytes. Moreover, the in vivo silencing of miR-326 could decrease the severity of autoimmune encephalomyelitis in mice as it was associated with fewer Th17 cells. MiRNA-binding site prediction software coupled with analysis of reporter activity of different 3′-UTR regions in the presence of miR-326 indicated that the *Ets-1* transcript could be a target of miR-326 [16]. *Ets-1* has been previously found to be a negative regulator of Th17 differentiation [17]. Thus, miR-326 overexpression might promote Th-17 differentiation by downregulating *Ets-1*. Another microRNA was also involved in Th17 differentiation. Indeed, in a mouse model of miR-155-deficient CD4 T cells, Escobar et al. observed a reduced secretion of IL-22 in Th17 lymphocytes, demonstrating the role of miR-155 in the expression of IL-22. MiR-155 binds to the *JARID2* mRNA and inhibits its translation. JARID2 is a transcriptional repressor which is responsible for the recruitment of the PRC2 complex (polycomb repressive complex 2) and mediates gene silencing through H3K27 trimethylation. In the absence of miR-155, JARID2 directly binds to the *IL-22* locus and is associated with the presence of the repressive histone mark H3K27me3, thus inhibiting *IL-22* transcription [18].

### 2.2. Th17 Plasticity

Multiple studies have witnessed Th17 cell conversion into other CD4 T cells. For instance, the in vitro and in vivo conversion of Th17 cells, in Th1 polarizing conditions, into a functional Th1 cell-like phenotype producing IFN-γ and lacking IL-17 secretion have been reported by Lee et al. [19]. The use of IL-17 reporter mice allow them to identify the extinction of IL-17 expression in Th17 cells simultaneously with an increase in IFN-γ production under IL-12 stimulation. The Th1 transcription factors STAT4 and Tbet induced by IL-12 and IL-23 acted together to convert Th17 progenitors into Th1-like cells. This phenomenon was also described in a model of colitis in which Th17 cells from IL-17F reporter mice were converted into IFN-γ-producing cells and were linked to disease development [20]. Other examples demonstrating Th17 cell high plasticity have been described such as the co-expression of two different lineage specific transcription factor at the same time in Th17 cells. These double-positive cell populations (like RORγt and GATA3 or RORγt and Foxp3) have functional properties and can express cytokines of both subsets [21,22]. GATA3 is the master transcription factors of the Th2 lineage whereas FoxP3 is mainly expressed by regulatory T cells [23,24]. GATA3+ RORγt+ T cells are able to produce both Th17 and Th2 cytokines: IL-17 and IL-4. The co-expression of RORγt and Foxp3 was linked to immunosuppressive function and these cells transiently expressed IL-17 at the beginning of the Th17 to Treg conversion [21]. Epigenetic regulation of the Th1 cell-like phenotype induced by the in vitro culture of Th17 cells with IL-12 was correlated with a decrease in permissive H3K4me modifications and histone acetylations at the *Il-17* locus and an increase in these marks at the IFN-γ locus [25].

Moreover, a global mapping of H3K4me3 and H3K27me3 marks in CD4 T cells revealed unexpected presence of both permissive (H3K4me3) and repressive (H3K27me3) histone marks on the promoter of lineage-specific transcription factors. The authors discovered a broad spectrum of chromatin states at these loci which may explain CD4 T cell plasticity. In addition, in Th17 cells, the Foxp3 promoter did not display repressive histone marks thus bringing evidence for Th17/Treg plasticity. Interestingly, specific cytokines of the different CD4 T cells lineage (like IL-17 for Th17 cells or IFN-γ for Th1 cells) were characterized by permissive histone modifications in a specific subset and the acquisition of repressive marks in the opposite lineages. For instance, the *Il17* and *Il21* loci were marked by H3K27me3 in Th1, Th2, and regulatory T cells and were associated with permissive H3K4me3 marks in Th17. These observations corroborated with the previous in vitro experiments and suggest that the signature of cytokines of a specific CD4 T cell subset is strongly repressed in the other subsets, but also bring insights to a more complex regulation of CD4 T cell lineage and interconversion possibilities [26,27]. Other arguments for the implication of epigenetics in the plasticity of the Th17 subset involved DNA methylation. Yang et al. performed genome-wide methylome analysis in ex vivo CD4 T cells and demonstrated that Th17 and naïve CD4 T cells had a similar methylation profile. Indeed, a higher number of demethylated regions was observed in Th17 and naïve CD4 T cells compared to Th1 cells. This absence of repression is in line with the plasticity of Th17 cells [28]. Other genome methylation analysis of Th17 versus Th1 cells expressing both IFN-γ and IL-17 (also called non-classic Th1 cells) provided evidence for the Th17 plasticity toward the Th1 phenotype and revealed that like Th17 and unlike classic Th1 cells, non-classic Th1 cells showed demethylation of the *RORC2* and *Il-17* promoters [29]. Despite that, the Th17 cells represent a specific CD4 T cells lineage, the plasticity of this subset to potentially convert into other CD4 T cells is now admitted and these plastic capacities are related to epigenetic modifications.

## 3. The Influence of the Tumor Microenvironment on the Epigenome of the Tumor Infiltrating Lymphocytes (TILs)

### 3.1. Pro-Tumoral Function of Th17 Cells 

A high level of tumor infiltrating Th17 cells can have opposite effects on survival depending on the type of cancer. On the one hand, in colorectal cancer, tumor infiltrating Th17 cells were correlated with lymph node metastases and affected negatively the postoperative survival [30]. In a cohort of 125 frozen colorectal tumor specimens, Tosolini et al. established that predominance of Th17 cells was associated with poor prognosis whereas patients with a high number of Th1 cells had prolonged disease-free survival [31]. On the other hand, in epithelial ovarian cancer, the predominance of Th17 cells over regulatory T cells represented an independent prognostic factor for overall patient survival [32]. 

Globally, the presence of intra-tumoral Th17 in colorectal cancer, hepatocellular carcinoma, gastric and pancreatic cancer was correlated with a bad prognosis whereas the accumulation of Th17 cells in malignant pleural effusion, prostate or epithelial ovarian cancer predicted improved patient survival [31,33,34,35,36,37]. The negative effects of Th17 cells on overall survival of some cancers may be in part explained by the role of IL-17. IL-17 produced by Th17 cells is believed to play a role in the promotion of cancers such as colorectal cancer or uterine cervical cancer by favoring an inflammatory environment propitious to cancer development and angiogenesis [38,39,40]. IL-17 is also believed to signal to colorectal tumor cells to dampen their production of the chemokines CXCL9/10, thus inhibiting CD8 T cells and Treg infiltration and promoting cancer progression [41]. The receptor of IL-17 present on tumor and tumor-associated stromal cells can induce the IL-6 production when activated leading to activation of the oncogenic factor STAT3. STAT3 signaling in tumor and stromal cells upregulated the expression of pro-survival and pro-angiogenic genes, and hence drove tumor growth [42]. Furthermore, Th17-mediated immunosuppressive activities may be linked to the expression of the ectonucleotidases CD39 and CD73 [43]. CD39 and CD73 convert extracellular adenosine triphosphate (ATP) into adenosine [44]. Extracellular ATP impacts a variety of immune cells. In CD8, ATP released in the environment is required for T cell receptor (TCR) activation [45], whereas in regulatory T cells, extracellular ATP promotes death signaling through the engagement of the purinergic P2X receptor [46]. ATP in the environment also favors dendritic cell (DC) maturation and macrophage M1 polarization [44]. Extracellular ATP depletion by Th17 cells expressing CD39 and CD73 thus inhibits inflammation by promoting subsequent Treg-mediated immunosuppression and diminution of CD8 T cell activation.

Th17 cells might also be responsible for the development of an immunosuppressive micro-environment through the recruitment of myeloid-derived suppressor cells (MDSCs) [47]. Indeed, in tumor bearing mice, administration of IL-17 was associated with a decrease of CD8 T cell infiltration and an increase of MDSCs in tumors. Furthermore, the number of MDSCs in the spleen was reduced by anti-IL-17 antibodies. The level of Matrix metallopeptidase 9 (MMP-9), which is produced by MDSCs and promotes tumor progression, was diminished in mice lacking the IL-17 receptor. Moreover, the study of He et al. revealed that IL-17 was required for the development and tumor promoting activity of MDSCs in tumor bearing mice. 

### 3.2. Anti-Tumoral Function of Th17 Cells

On the contrary, the adoptive transfer of tumor-specific Th17 cells in a mouse model of lung melanoma prevented tumor growth linked to the activation of tumor-specific CD8 T cells and promoted DC infiltration in the tumor [48,49]. The adoptive transfer of tumor-specific Th1 cells did not have any curative effect. Since the administration of Th17 cells in CCR6-deficient mice did not inhibit tumor growth, the authors hypothesized that CCR6 is required for the response to Th17 therapy. Moreover, CCR6 was expressed on a specific type of DCs (CD8α+ DCs) whose number was enhanced by the injection of Th17 cells. The protective function of Th17 cells against tumors may be due to their ability to enhance inflammatory responses through the CCR6-CCL20 axis and results in increased antigen presentation by dendritic cells. Another anti-tumoral effect of Th17 cells was also observed by Nuñez’s laboratory who demonstrated that the inhibition of Th17 cell recruitment in the tumor in a model of RORγt-deficient mice was associated with an increased tumor growth which could be reverted by an adoptive transfer of Th17 cells [50]. The plasticity of Th17 into the Th1 phenotype could also explain Th17 cell anti-tumoral properties with the benefit of IFN-γ production on anti-tumor immunity. In all in vivo studies mentioned above, we must keep in mind that Th17 are not the only cells producing IL-17. Indeed, innate lymphoid cells (ILC), the innate homologues of T helper subsets, can produce cytokines such as IL-17 and evidences for their involvement in tumor immunity are increasing day by day [51,52]. The controversial role of IL-17 and Th17 in cancer is further developed in different reviews [53,54]. Briefly, IL-17 produced by Th17 cells can stimulate angiogenesis and promote the formation of metastasis by inducing the expression of metalloproteinases in cancer cells thus altering the extracellular matrix and favoring tumor invasion. In contrast, Il-17 can promote the activation and recruitment of anti-tumoral immune cells, including macrophages, neutrophils, natural killer cells (NK), and cytotoxic CD8 T cells.

Beside these observations, the bivalent role of Th17 cells in cancer may be due to heterogeneity in cytokine production depending on the environmental context. In vitro experiments showed that mouse Th17 cells polarized without TGF-β (IL-6 + IL-1β + IL-23) did not produce the immunosuppressive cytokine IL-10 and expressed the Th1 cell-related proteins such as the transcription factor Tbx21 and IFN-γ [55]. The mechanisms involved in the conversion of prototypical Th17 cells into cells secreting more inflammatory or more immunosuppressive cytokines will need to be further investigated. Tanaka et al. reported that the tripartite motif-containing 33 protein (Trim33), a modulator of TGF-β signaling seemed to play a role in the pro-inflammatory activity of Th17 cells [56]. Moreover, the complexity of IL-17 signaling has also to be taken in consideration to explain the bivalent role of Th17 in cancer.

Whether the presence of Th17 is associated with a good or bad prognosis, the modality by which Th17 cells accumulate within the tumor has not been elucidated yet. Over-representation of Th17 cells could result of the recruitment and the expansion of peripheral Th17 cells, a differentiation process from naive precursors or the plastic conversion of other CD4 T cells into Th17 cells. The impairment of recruitment, differentiation, or expansion of other CD4 T cells subsets might also be responsible for the intra-tumoral predominance of Th17 lymphocytes. Regarding the importance that epigenetics plays in CD4 T cell differentiation and plasticity, especially in Th17 cells, we can wonder to what extent epigenetics affect or may explain the Th17 intra-tumoral predominance in the cancer types previously mentioned. 

## 4. Epigenetic Factors Leading to Th17 Cell Predominance in the TME (Tumor Micro-Environment)

During the last decade, studies on the epigenetic interplay between immune, stromal and cancer cells in the tumor microenvironment have emerged. These publications were particularly focused on the impact of epigenetics on the formation of CAFs (cancer associated fibroblasts) and CD8 T cell exhaustion (TOX) but more data on CD4 T cell biology need to be collected [57,58,59]. 

### 4.1. The Cytokines Produced by the TME Drive the Th17 Polarization and Expansion

One of the mechanisms which could explain Th17 cell prevalence among TILs is a direct polarization of CD4 T cells into Th17 under the influence of cytokines produced by the tumor microenvironment. Su et al. reported that tumor cells and tumor-derived fibroblasts derived from melanoma, colon cancer, and breast cancer, secreted cytokines required for the generation of human Th17 cells (IL-1β, Il-6, IL-23, and TGF-β) [60] (Figure 2a). Rezalotfi et al. also reviewed the role of gastric cancer stem cells on Th17/Treg balance. They indicated that the cytokines secreted by these cells (IL-1β, Il-6, IL-8, IL-23 and TGF-β) could explain the high infiltration of Th17 cells at tumor site [61]. The in vitro experiments carried out by Ye et al. showed that IL-1β, IL-6, IL-23 present in malignant pleural effusion could promote Th17 cell differentiation from naive CD4 T cells [36]. In addition, Qian et al. linked the overexpression of IL-23 in tumor tissues isolated from mice and human breast cancer patients with an increased Th17 representation [62]. As described earlier, epigenetics is known to play a key role in the cytokine-driven initiation of Th17 differentiation along with the recruitment of histones and DNA modifying complexes at the promoters of Th17-specific genes, including *IL-17* and *RORγt* leading to their expression. These epigenetic enzymes are recruited by specific transcription factors (STAT3) or other epigenetic regulators (TRIM28, Cxxc1, Ikaros) which are themselves activated upon TCR and cytokine signaling (activation of the receptors of IL-6, TGF-β, IL-1β, and IL-23). 

### 4.2. Epigenetics May Enhance Th17 Recruitment or Inhibit the Recruitment of Other CD4 T Cells at the Tumor Site

Yu et al. reported that the accumulation of Th17 cells in cervical cancer was positively associated with a high level of CCL20 in tumor tissues. CCR6 is expressed at the surface of Th17 cells and is the receptor of the chemokine CCL20. Hence, the CCR6-CCL20 pathway may play a role in the recruitment of Th17 cells at tumor site [63,64] (Figure 2b). Recently, the expression of CCL20 in colorectal cancer was found to be regulated by a novel long non-coding RNA named u50535. Like miRNAs, long noncoding RNAs (lncRNAs) are part of the noncoding RNA group but contain more than 200 pb. LncRNA microarray revealed that the lncRNA-u50535 was remarkably upregulated in colorectal cancer associated with poor prognosis. To investigate the genes regulated by lncRNA-u50535, Yu et al. performed RNAseq assays when this lncRNA was either inhibited or overexpressed. Their results showed that the lncRNA-u50535 upregulated *CCL20* expression and that it is the upstream regulator of this chemokine. Besides the direct recruitment of Th17 cells through the CCR6-CCL20 axis, Th17 over-representation in tumors could be the result of the inhibition of the recruitment of other CD4 T cell subsets. The chemokines CXCL9 and CXCL10 attract Th1 cells thanks to their receptor CXCR3. [63,65,66,67]. Thus, in human models of colon and ovarian cancer, Peng et al. as well as Nagarsheth et al. demonstrated that enhancer of zeste homologue 2 (EZH2), which mediates the repressive histone mark H3K27me3, inhibited the expression of the Th1-type chemokines CXCL9 and CXCL10. In ovarian cancer, DNA methyltransferase 1 (DNMT1), which is responsible for DNA methylation, also regulated the expression of these chemokines. These two epigenetic enzymes thus prevented Th1 cell recruitment within the tumor site and were associated with poor CD8 T cell infiltration and prognosis (Figure 2c). These studies highlighted the importance of DNA and histone methylation in the epigenetic regulation of the main antitumor immune cells, including Th1 and CD8 T cells, into the tumor microenvironment [68,69]. 

### 4.3. Modification of the Epigenome by the Tumor Micro-Environment

#### 4.3.1. The Hypoxic Tumor Microenvironment

Johnson et al. suggested that hypoxia may induce a remodeling of histone marks in tumor cells such as an increase of the permissive mark H3K4me3 at promoters of genes which regulate the hypoxic stress or a decrease of the repressive mark H3K27me3 at those promoters in tumor cells [70,71]. The in vitro culture of Th17 cells under hypoxia strongly induced the hypoxia-inducible factor 1 alpha (HIF-1α) and dramatically affected cytokine production compared to Th17 cells polarized under ambient oxygen. Surprisingly, the authors found that hypoxia led to the secretion of IL-10, an immunosuppressive cytokine, by Th17. Although hypoxia induced the expression of a different cytokine profile in Th17 and revealed plasticity in vivo, these observations have not been observed in the tumor and have not been related to chromatin remodeling yet [72]. 

#### 4.3.2. Metabolism

Cancer cells are characterized by multiple metabolic aberrations which deregulate the homeostasis and cause accumulation of metabolites. Many cofactors of epigenetic enzymes that modify DNA methylation status or histone post-translational modifications are derived from intermediates of cellular metabolic pathways. Modifications of the epigenome not only occur in cancer cells but since metabolites are released in the environment, they will also affect the surrounding cells. Cancer is thus affecting the deposition of regulatory marks and chromatin accessibility in the tumor micro-environment. As an example, DNA and histone methyltransferases require the metabolite S-adenosylmethionine (SAM) as a methyl donor. After the donation of a methyl group, SAM is then transformed into the metabolite S-adenosylhomocysteine (SAH). SAH can be recycled into SAM via hydrolysis to homocysteine. Alternatively, homocysteine can be catabolized to give taurine, cysteine and sulfate (Figure 3). The SAM/SAH ratio can be quantitatively modulated by the metabolic flux of the methionine cycle, thus affecting DNMT and HMT activity. An excess supply of SAM might for example contribute to DNA hypermethylation at CpG sites and inappropriate gene silencing [73].

Moreover, other epigenetic enzymes are linked to metabolism: HDAC classes I, II, and IV need acetyl CoA, which is a metabolite at the crossroads of several metabolic pathways, whereas HDAC III proteins, also called sirtuins, require NAD+ as a co-substrate for histone modification [74,75,76]. Sirtuin proteins are also responsible for the deacetylation of non-histone proteins such as the transcription factor STAT3. STAT3 deacetylation impedes its translocation to the nucleus and inhibits its binding to the *Rorc* promoter, thus impairing Th17 differentiation. Sirtuin-1 activation with the pharmacological agonist metformin was found to promote STAT3 deacetylation and inhibit Th17 cell differentiation in human and mouse CD4 T cells in vitro. Hence, the modification of the activity of sirtuins and other epigenetic enzymes can interfere with Th17 cell differentiation [77]. 

This metabolic regulation of gene expression can therefore affect Th17 prevalence (Figure 2d). Xu et al. demonstrated that Th17 could be converted into Treg cells by manipulating a single step of the glutamate metabolic pathway. The reduction of 2-hydroxyglutarate levels in Th17 cells induces a reduction of the *Foxp3* promoter methylation status leading to its expression and consequently to the inhibition of the master regulator of Th17 differentiation, RORγt [78].

In gastrointestinal cancer, gut microbiota dysbiosis has been linked to the development of cancer and may be an important environmental factor that can modulate the host epigenome [79]. For instance, dietary metabolites derived from gut microbiota can be critical regulators of epigenetic enzymes. Inhibition of HDAC by short-chain fatty acids produced by commensal bacteria can enhance histone acetylation of the *Foxp3* locus thus increasing its expression and Treg differentiation [80]. Fisetin, a flavonoid-derived metabolite produced by gut microbiota was found to reduce histone acetylation and inhibit the expression of the proinflammatory cytokines IL-6 and TNF-α [81]. Thus, the modification of the host epigenome by gut microbiota-derived metabolites can modulate CD4 T cells differentiation. The resulting imbalance between Th17 and Treg cells in the tumor microenvironment can promote cancer development by either creating a proinflammatory context (Th17 cells) or inhibiting anti-tumor responses (Treg).

### 4.4. Th17 Predominance in Tumor Can Be Modulated by Plasticity

#### 4.4.1. Treg to Th17 Cell Plasticity

In colorectal and esophageal cancers, the presence of hybrid Th17/Treg cells expressing both IL-17, RORγt and Foxp3 were identified by flow cytometry in the tumor and in the peripheral blood of patients. These hybrid cells also expressed CCR6 and CXCR3. In colorectal cancer tissues, western blot analyses revealed that CXCL11 expression, known to be the ligand of CXCR3, was enhanced compared to normal tissues. Furthermore, hypoxia was found to increase CXCL11 expression in macrophages associated with colorectal cancer (CD68+ cells). Chemotaxis tests performed in vitro using a transwell system confirmed that Tregs could be attracted by CXCL11. Since the in vitro culture of CD4^+^ CD25^high^ regulatory T cells under hypoxia also induced the expression of IL-17, the authors suggested that CRC-derived CD68+ cells attracted Foxp3+ Tregs at the tumor site where hypoxia induced their expression of IL-17. The negative effect of Foxp3+ IL-17+ T cells on cancer prognosis may be due to the fact that these cells promote cancer growth. Indeed, co-culture of Foxp3+ IL-17+ T cells with colorectal cancer cells increased the number of cancer cell colonies [82,83]. Other studies discovered that a small fraction of these hybrid Th17/Treg cells was found in the peripheral blood of healthy humans and were selectively accumulated in the colitis inflammatory micro-environment of ulcerative patients. These Th17/Treg cells display inflammatory properties with the secretion of IL-1 β and IL-6 which can be linked to the pathogenesis of this disease. In vitro, Foxp3+ IL17+ T cells also demonstrated similar suppression of CD8 T cell proliferation and IFN-γ production. Since ulcerative colitis is often associated to the development of cancer, the pro-inflammatory environment produced by Th17 cytokines combined with the inhibition of local T cell immunity (suppressive capacities of Tregs) may mechanistically link human chronic inflammation to tumor development [84]. In vitro, Foxp3+ IL17+ T cells were generated from CD4^+^ CD25^high^ Treg cells in the presence of TGFβ, IL-2, and myeloid antigen presenting cells. Li et al. also investigated the mechanisms underlying the conversion of CD4+ CD25- Tregs into IL-17-producing cells in the presence of antigen presenting cells, IL-1 β, and IL-12 and discovered that the transcription factors Runx1 and Runx3 were critical regulators of the generation of these cells. Knock-down of Runx1 expression significantly reduced the expression of Rorγc and Foxp3, suggesting that Runx1 is involved in the production of IL-17 in Tregs. Higher levels of Runx3 protein were also found in CD4+ CD25- Tregs producing IL-17 compared to IL-17-Tregs [85,86]. Interestingly, phenotypic and functional characterization of Th17 TILs from melanoma, ovarian, breast and colon cancers by FACS analysis revealed the possibility of the conversion of Th17 cells into Foxp3 + T cells (Figure 2e) [87]. Following several expansion cycles of Th17 TILs in the presence of irradiated allogenic PBMCs and TCR stimulation with OKT3, the percentage of IL-17 producing Th17 cells dropped remarkably whereas the percentage of Foxp3+ and CD25 + T cells increased amongst the Th17 clones. Interestingly, RT-PCR assays performed using TCR-V β -specific primers, before and after expansion of the Th17 TILs, did not show significant differences, suggesting that the homogeneity of the clonality was preserved. In conventional Tregs, the *Foxp3* promoter is demethylated compared to other CD4 T cells, thus allowing a stable expression of Foxp3. In expanded Th17 cells, the authors found that the multiple cycles of in vitro TCR stimulation resulted in a decrease of the methylation of the *Foxp3* promoter, indicating an epigenetic regulation responsible for the Th17 cell conversion into Tregs [87].

#### 4.4.2. Th17 to Th1 Cell Plasticity 

In addition to Foxp3 expression in the Th17 clones of TILs, Ye et al. found that repeated in vitro TCR stimulation with OKT3, PBMCs and IL-2 could also increase the production of IFN-γ by these Th17 cells. Upon in vitro activation of TILs, the expression levels of RORγt and IRF-4 (specific of Th17 cells) were dramatically diminished and the expression of T-Bet (master regulator of Th1 cells differentiation) in the expanded Th17 clones significantly increased with stimulation and expansion. [87]. In vivo, the same observation was assessed using FACS analysis of CD4 T cells specific of the tumor antigen MAGE-A3. Among MAGE-A3 specific helper T cells, some of them were found to express both IFN-γ and IL-17. Investigation of the fine specificity of peptide recognition revealed that these cells originated from the same clone. Th1 and Th17 MAGE-A3 specific T cells were overserved respectively at late (effector memory CCR7- cells) and early differentiation stages (central memory CCR7+ cells) based on the expression of CCR7. Hamaï et al. thus witnessed the possible conversion of Th17 cells into Th1/Th17 cells producing IFN-γ [88]. Hence, Th17 cells are characterized by an unstable lineage phenotype and can display differential plasticity after TCR stimulation. However, the modality of CD4 T cell plasticity in the context of cancer is not yet totally understood. Whether Th17 cells will convert into other CD4 T cells at the tumor site or whether they will convert at the periphery and will then be recruited within the tumor is unknown. 

A summary of the different factors that could modulate the epigenome of Th17 cells in the context of cancer and further affect their differentiation, recruitment or plasticity is provided in Table 1. In cancers where Th17 cell infiltration is associated with a good prognosis, it will be important to understand and control the epigenetic mechanisms regulating Th17 cell predominance to shape and conserve a favorable tumor microenvironment. In epithelial ovarian cancer, a high Treg/Th17 ratio was associated with tumor progression. Zhou et al. showed that exosomes derived from tumor-associated macrophages (TAMs) were enriched in microRNAs which could directly inhibit STAT3 and drive Treg differentiation into CD4 T cells, thus creating an immunosuppressive microenvironment that facilitates cancer progression and metastasis spreading [32]. 

Th17 cell prevalence amongst TILs may be the result of a combination of the different mechanisms listed above (polarization and expansion at the tumor site, recruitment, conversion into other CD4 T cell subsets). However, whether Th17 cells undergo a conservation of the epigenetic imprinting characteristic of these cells or a remodeling of the epigenome by the tumor micro-environment remains unknown. The previous possibilities listed above need to be further investigated. Now, the tracking of the accessible chromatin landscape at the single cell level is possible thanks the development of the single-cell ATAC-seq technology (assay for transposase-accessible chromatin). This technique allows for the identification of relaxed chromatin portions thanks to a transposase which binds to these regions. Open chromatin is favorable to transcription and the epigenome of an individual cell is investigated with the use of an individual barcoded sequence for each cell [89]. 

The role played by epigenetics in the understanding of Th17 intra-tumoral predominance may also provide other hypothesis in terms of therapies to get rid of the negative effects which Th17 cells might have in certain types of cancer. 

## 5. The Promising Use of Epidrugs in Combination with Other Therapies for Anti-Tumor Treatment

An effective anti-tumor immunity requires the collaboration of both CD4 and CD8 T cells [90]. As we saw previously, the type of CD4 T cells infiltrating the tumor can affect dramatically cancer prognosis and can be in part regulated by epigenetic mechanisms. In this part, we are talking about the recent advances in cancer treatment regarding the molecules targeting epigenetic enzymes called epidrugs. 

### 5.1. Combination of Immune Checkpoint Blockade with Epidrugs 

For the past few years, immunotherapies have revolutionized the treatment of cancer and have been used to prevent immune escape with checkpoint blockade inhibitors like anti-programmed cell death protein-1 (PD-1), anti-programmed cell death ligand-1 (PD-L1) or anti-cytotoxic T-lymphocyte-associated protein 4 (CTLA-4) antibodies. However, surprisingly, only few patients benefit from these treatments since immune checkpoints are frequently regulated by epigenetics within tumors leading to immune resistance [91].

The use of the class I-selective HDAC inhibitor Domatinostat to enhance the expression of PD-1 on immune cells or PD-L1 on tumor cells and then to potentiate the efficiency of immunotherapies has been investigated by Bretz et al. and has shown promising results in mice with an increase of lymphocyte infiltration and tumor regression. This study further supports the development of epigenetic therapy in combination with existing cancer immunotherapy [92]. Another study showed that the inhibition of LSD1 (histone lysine-specific demethylase) associated with an anti-PD-1 antibody suppressed tumor growth and lung metastases in breast cancer. This mechanism enhanced the expression of chemokines CCL5, CXCL9, CXCL10 and the recruitment of cytotoxic T cells leading to the promotion of intra-tumoral CD8 T cell infiltration in mice bearing triple-negative-breast cancer xenograft tumors [93]. 

The combination of an EZH2 inhibitor (DZNep) together with a DNMT1 inhibitor (5-Azacytidine) and an anti-PDL-1 antibody for the treatment of Hepatocellular Carcinoma showed promising results and will be deeply investigated in further clinical trials. Indeed, EZH2 and DNMT1 are responsible for the repression of the expression of multiple genes including tumor antigens and the Th1-attracting chemokines CXCL9 and CXCL10, thus inhibiting an efficient anti-tumor response. RT-PCR analyses revealed that the inhibition of the expression of these enzymes in association with the immunotherapy led to an increased expression of *CXCL9* and *CXCL10* mRNAs which was correlated in vivo with an enhanced recruitment of cytotoxic T-cells at the tumor site. Moreover, the cancer testis antigens NY-ESO-1 and LAGE were induced upon treatment. In vivo, these observations were associated with a significant tumor regression when the mice were treated with the combination of epigenetic therapy and immunotherapy [94]. 

Recently Khan et al. discovered that the transcription factor TOX is responsible for CD8 T cell exhaustion and induces PD-1 expression [59]. As TOX seems to regulate epigenetic accessibility by altering a network of transcription factors interfering with specific genes of exhausted T cells, the development of molecules targeting TOX may represent new strategies to prevent cytotoxic T cell exhaustion and immune escape. 

### 5.2. Favoring Tumor Immunogenicity with Epidrugs 

Currently, epigenetic therapies are used in leukemia treatment (with the hypomethylating agent 5-Azacytidine) but are not in solid tumors. Multiple studies of potential combination of epidrugs with conventional therapies have emerged, mostly to target cancer cells by downregulating proto-oncogene expression and re-expressing tumor suppressor genes or to increase the recruitment of specific CTLs in the tumor site by expressing de novo antigens like ERV (endogenous retroviruses) [95]. Indeed, in human genome, DNA methylation is responsible for ERV silencing. DNMT inhibitors can induce the re-expression of ERV and can lead to the recognition of these neo-antigens by CTLs and the production of IFN-γ [96]. Other strategies have been developed to enhance antigen presentation using histone deacetylase inhibitors to increase MHC class I expression and facilitate tumor antigen presentation [97,98]. 

### 5.3. Remodeling of the Tumor Micro-Environment by Epidrug Treatment

Epidrugs can also be used to remodel the tumor micro-environment and suppress immunosuppressive cells. For instance, the class I HDAC inhibitor Entinostat neutralizes regulatory T cells in vitro, in renal and cancer models, hence enhancing efficacy of immunotherapies [99]. In vivo, in two syngenic mouse tumor models, Entinostat could alter the function of myeloid-derived suppressor cells by inhibiting the expression of the immunosuppressor COX-2, arginase-1, iNOS proteins, and other key cytokines/chemokines [100]. 

## 6. Epigenetic Reprogramming of CD4 T Cells to Modulate Th17 Cell Differentiation and Plasticity

The Th17 subset is characterized by an unstable lineage phenotype and can display differential plastic capacities and cytokine profiles and it is clear now that Th17 differentiation and plasticity programs are modulated by epigenetics. In line with these previous observations, it has been considered to study the development of epigenetic therapies to impair Th17 differentiation, to limit its plastic capacities or to favor the expansion of these cells in cancer when it is associated with a good prognosis. For example, inhibition of JMJD3 in CD4 T cells (*Jmjd3* KOc) was described by Zou et al. to increase the percentage of Th17 and Th2 cells in the small intestine and colon in mice. JMJD3 catalyzes the demethylation of H3K27 as well as the demethylation of H3K4, which is a permissive mark. Moreover, *Jmjd3* ablation in CD4 T cells decreased the expression of the Th1 specific transcription factor *T-bet*, thus impairing Th1 differentiation [101]. The regulation of the Th17/Treg balance can also be modulated by epidrugs. In a mouse model of allograft rejection, the inhibition of the deacetylase Sirtuin 1 by sirtinol reduced the proportion of Th17 cells and inhibited their functional properties by decreasing the expression of IL-17A and RORγt. Sirtinol also enhanced Foxp3 expression and allograft tolerance [102]. The expression of the *Foxp3* gene is dependent on its acetylation status. The inhibition of Sirtuin-1 increased *Foxp3* acetylation and expression and the formation of functional Treg cells. On the contrary, deacetylation of RORγt by Sirtuin-1 enhanced its transcriptional activity and favored the differentiation of Th17 cells. SIRT1 inhibitors suppressed Th17 differentiation and were protective in a mouse model of multiple sclerosis and IBD [103,104,105]. 

Besides Sirtinol, SAHA (Suberoylanilide hydroxamic acid), a histone deacetylase inhibitor (HDACi), can also regulate the Th17/Treg balance. In a mouse model of allograft rejection, a treatment with SAHA could increase the survival of the recipient mice and it was shown that these immunosuppressive functions were linked to a lower level of IL-17 expression. SAHA differently impacted Foxp3+ T cells depending on the dose of SAHA used. High doses of SAHA (0.5 and 1 μM) significantly suppressed the generation of these cells whereas low doses (0.1 μM) of SAHA enhanced the proportion of Treg [106].

## 7. Conclusions

The presence of Th17 cells in the tumor could either be beneficial or detrimental depending on the cancer type. This cell population displays great plasticity and can convert into other CD4 T cells thanks to epigenetic modifications. We reviewed above the different factors which could explain Th17 cell predominance in TME including cytokine-driven polarization, and conversion of Th17 cells into other CD4 T cell subtypes. We described that epigenetics could alter immune infiltration with the regulation of the expression of Th1-attracting chemokines and determine T cell recruitment at the tumor site. Epigenetics therefore plays a critical role in immune evasion but the alteration of the metabolism due to cancer and the modification of the level of oxygen in the TME may also modify the epigenome of the TILs. The promising perspectives of epigenetic therapies to modify a favorable TME and prevent a Th17 cell landscape within the tumors where it is associated with a bad prognosis could then inhibit immune escape and improve patient’s outcome. A better understanding of the immune process mediated by epigenetics is the key to develop new strategies. 

## Figures and Tables

**Figure 1 ijms-21-01673-f001:**
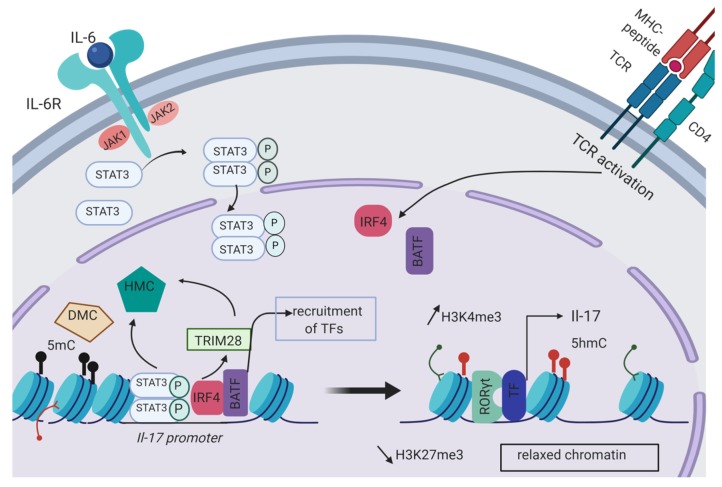
Epigenetic regulation of *Il-17* expression occurring during Th17 cell differentiation. The transcription factors IRF4 and BATF are induced upon TCR activation in CD4 T cells. IL-6 promotes STAT3 phosphorylation and dimerization through its receptor (IL-6R). Following nuclear translocation, STAT3, IRF4 and BATF bind to the *Il-17* promoter. Histone modification complexes (HMCs) including HMTs or HATs as well as DNA modification complexes (DMCs) like DNMTs are then directly recruited at the *Il-17* promoter by STAT3 or indirectly through the epigenetic regulator TRIM28. HMCs and DMCs are responsible for the formation of permissive histone marks like H3K4me3 or demethylation of CpG islands with the formation of 5-hydroxymethylcytosine (5hmC). On the contrary, DNA methylation (5mC) and repressive histone marks (H3K27me3) are decreased at the *Il-17* locus thus allowing chromatin remodeling and accessibility of the *Il-17* promoter to other transcription factors. Among the transcription factors required for Il-17 expression, RORγt is recruited to the *Il-17* promoter by TRIM28. Created with BioRender.com.

**Figure 2 ijms-21-01673-f002:**
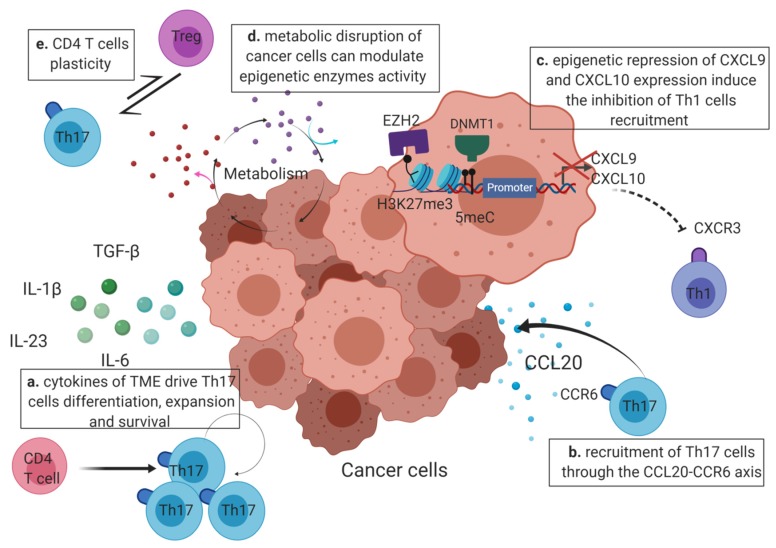
Factors regulating Th17 infiltration in the TME (tumor micro-environment). Th17-polarizing cytokines, like TGF-β, IL-6, IL-1β, and IL-23, produced in the TME are responsible for Th17 differentiation, proliferation and survival (bold arrow) (a). Tumor specific Th17 cells already differentiated may also be recruited through the CCL20-CCR6 axis (curved arrow) where the chemokine CCL20 produced by tumor cells binds to the Th17 receptor CCR6 (b). Th17 predominance in the TME may also be explained by the inhibition of the recruitment of Th1 cells (dotted lines). Indeed, in cancer cells, EZH2 and DNMT1 can lead to the epigenetic repression of the Th1-attracting chemokines CXCL9 and CXCL10 by the apposition of the repressive histone marks (H3K27me3) and DNA methylation at the *CXCL9* and *CXCL10* promoters (c). The metabolic dysregulation occurring in cancer cells generate the release of metabolites in the environment that might further interfere with the activity of epigenetic enzymes in CD4 T cells and favor the expression of genes required for Th17 differentiation or functionality (d). Overall, the balance between Th17 and other CD4 T cells, like Tregs, can be modulated by the plastic capacities of these immune cells and induce the prevalence of Th17 cells at the tumor site. The possible conversion between Th17 and other CD4 T cells is represented by double arrows (e). Created with BioRender.com.

**Figure 3 ijms-21-01673-f003:**
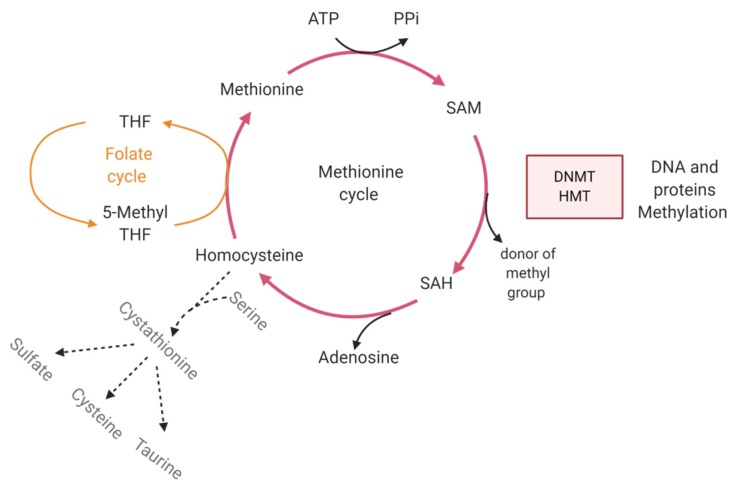
The methionine cycle and alternative homocysteine degradation regulate DNMT and HMT activity. HMT and DMMT catalyze histone and DNA methylation, respectively. The donor of the methyl group is the metabolite S-adenosylmethionine (SAM) which is then converted into *S*-adenosylhomocysteine (SAH). SAH is regenerated into SAM during the methionine cycle (red lines) where the loss of adenosine induces its transformation into homocysteine. The methyl group of 5-Methyltetrahydrofolate (5-MethylTHF) from the folate cycle (orange lines) is then transferred to homocysteine to form methionine. Depending of the metabolic needs, homocysteine can be catabolized into sulfate, taurine or cysteine. This alternative pathway is represented by dotted lines. The regeneration of SAH into SAM is then compromised and negatively impacts HMT and DNMT activity. Created with BioRender.com.

**Table 1 ijms-21-01673-t001:** Factors modulating the epigenetic regulation of genes involved in Th17 differentiation, recruitment or plasticity in the context of cancer.

	Genes	Epigenetic Regulation	Factors Modulating the Epigenome	References
Th17 differentiation	IL-17, RORγt	Histone modification complex like HAT or HMT responsible for the deposition of permissive marks	Cytokines of the TME	[9,10,12,13,14,15,16,17,60,61,62,74,75,76,77,80,81,82]
		DNA demethylation enzymes (5hmc)	Metabolism regulates the biodisponibility of co-factors of epigenetic enzymes	
		miRNA	Metabolites produced by gut microbiota modulate epigenetic enzymes activity	
Th17 recruitment through the CCR6-CCL20 axis	CCL20	long non coding RNA (lncRNA-u50535)	Upregulation of lncRNA-u50535 in colorectal cancer	[64]
Th17 plasticity				[74,75,76,77,79,88]
- Th17 /Treg plasticity	Foxp3	DNA methylation	Glutamate Metabolism pathway (reduction of 2 hydroxyglutarate level in Th17 cells diminish the *Foxp3* promoter methylation status)	
- Th17/Th1 cells plasticity	IFN-γ, Tbet	Histone modification complex	Exposition of differentiated Th17 cells to another cytokinic environment	

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
