# Peer review of "The Fate of Th17 Cells is Shaped by Epigenetic Modifications and Remodeled by the Tumor Microenvironment"

_ijms, 2020, doi:10.3390/ijms21051673_

Round 1

Reviewer 1 Report

Renaude and colleagues in their review titled “The fate of Th17 cells is shaped by Epigenetic modifications and remodeled by the tumor microenvironment” reviewed the role of Th17 cells a subset of CD4+ T cells characterized by the master transcription factor RORγt and the production of IL-17 and the role of epigenetic modifications such as post-translational histone modifications and DNA methylation in Th17 cell differentiation and high plasticity. Tey reviewed the role played by Th17 cells in different types of cancers its association with prognosis. They have also nicely reviewed the remodeling of the epigenome by the tumor microenvironment explaining Th17 cell predominance. They have concluded the review by discussing the promising treatment perspectives of molecules targeting epigenetic enzymes to remodel a Th17-enriched tumor microenvironment.

All in all, it is an excellent review with all the literature and I don’t have further comments except very minor.

The review “Gastric Cancer Stem Cells Effect on Th17/Treg Balance; A Bench to Beside Perspective” is relevant and may be added to this review.

And also the authors may add any information that the microbiome has on tumor microenvironment and its influence on epigenetic changes.

Reviewer 2 Report

In this manuscript the authors review in detail the role of Th17 cell infiltration in cancer (including several tumor models). They put in evidence the plasticity of these cells, which can change into other phenotype of T-cells, in part regulated by epigenetic mechanisms. Overall, authors describe the influence of epigenetics in regulating the tumor microenvironment and review how to possibly enhance immune therapies by combination with epidrugs.

The manuscript is well organized, and well written, language is clear and the topic is pertinent. I have minor suggestions of improvement only:

  • my main suggestion is that authors could provide a summary table with the most important findings they describe that actually link epigenetic phenomena with regulation of Th17 cells and other TILs. Authors describe a lot of data from many papers throughout the text, and despite being well-organized, a summary Table would be beneficial for the reader to follow the manuscript.
  • there are some typos, for instance: Figure 3d "can modulate" instead of "can modulates"; line 470 "recent advances" instead of "recent advanced"; line 166 "factor" instead of "factors".
  • the sentence starting on line 190 is too large and difficult to follow, and could be made simpler.
  • line 304: authors mean "long non-coding" instead of "long coding".
